# Pharmacokinetic Model Analysis of Supralingual, Oral and Intravenous Deliveries of Mycophenolic Acid

**DOI:** 10.3390/pharmaceutics13040574

**Published:** 2021-04-17

**Authors:** Xiuqing Gao, Lei Wu, Robert Y. L. Tsai, Jing Ma, Xiaohua Liu, Diana S.-L. Chow, Dong Liang, Huan Xie

**Affiliations:** 1Department of Pharmaceutical Science, College of Pharmacy and Health Sciences, Texas Southern University, Houston, TX 77004, USA; x.gao3704@student.tsu.edu (X.G.); jing.ma@tsu.edu (J.M.); dong.liang@tsu.edu (D.L.); 2Department of Pharmcological & Pharmaceutical Sciences, College of Pharmacy, University of Houston, Houston, TX 77204, USA; lwu8@central.uh.edu (L.W.); phar33@central.uh.edu (D.S.-L.C.); 3Department of Translational Medical Sciences, Institute of Biosciences and Technology, Texas A&M Health Science Center, Houston, TX 77030, USA; robtsai@tamu.edu; 4Department of Biomedical Sciences, Baylor College of Dentistry, Dallas, TX 75246, USA; xliu1@tamu.edu

**Keywords:** mycophenolic acid, supralingual, pharmacokinetic modeling, intravenous, oral, transmucosal drug delivery, enterohepatic recycling

## Abstract

Mycophenolic acid (MPA) is commonly used for organ rejection prophylaxis via oral administration in the clinic. Recent studies have shown that MPA also has anticancer activities. To explore new therapeutic options for oral precancerous/cancerous lesions, MPA was designed to release topically on the dorsal tongue surface via a mucoadhesive patch. The objective of this study was to establish the pharmacokinetic (PK) and tongue tissue distribution of mucoadhesive MPA patch formulation after supralingual administration in rats and also compare the PK differences between oral, intravenous, and supralingual administration of MPA. Blood samples were collected from Sprague Dawley rats before and after a single intravenous bolus injection, a single oral dose, or a mucoadhesive patch administration on the dorsal tongue surface for 4 h, all with a dose of 0.5 mg/kg of MPA. Plots of MPA plasma concentration versus time were obtained. As multiple peaks were found in all three curves, the enterohepatic recycling (EHR) model in the Phoenix software was adapted to describe their PK parameters with an individual PK analysis method. The mean half-lives of intravenous and oral administrations were 10.5 h and 7.4 h, respectively. The estimated bioavailability after oral and supralingual administration was 72.4% and 7.6%, respectively. There was a 0.5 h lag-time presented after supralingual administration. The results suggest that the systemic plasma MPA concentrations were much lower in rats receiving supralingual administration compared to those receiving doses from the other two routes, and the amount of MPA accumulated in the tongue after patch application showed a sustained drug release pattern. Studies on the dynamic of drug retention in the tongue after supralingual administration showed that ~3.8% of the dose was accumulated inside of tongue right after the patch removal, ~0.11% of the dose remained after 20 h, and ~20.6% of MPA was not released from the patches 4 h after application. The data demonstrate that supralingual application of an MPA patch can deliver a high amount of drug at the site of administration with little systemic circulation exposure, hence lowering the potential gastrointestinal side effects associated with oral administration. Thus, supralingual administration is a potential alternative route for treating oral lesions.

## 1. Introduction

Oral cancer occurred in 355,000 people globally in 2018 and resulted in nearly 50% death in patients diagnosed with this disease [1]. Oral squamous cell carcinoma (OSCC) constitutes the majority (>90%) of oral cancers. Evidence has suggested that OSCC may have developed from precancerous lesions, such as leukoplakia and lichen planus [2]. Current oral cancer treatment modalities include surgery, radiation, and chemotherapy, depending on tumor location and stage [3]. However, serious side effects are often seen with systemic chemotherapy and surgical intervention, presenting major challenges to clinical practice [4]. A reasonable strategy to reduce cancer-related mortality and morbidity is to develop non-invasive methods to treat precancerous lesions or early-stage cancers [5].

Oral mucosa overlining the tongue and the buccal surface is a route for drug administration that bypasses the first-pass effect, avoids pre-systemic metabolism, and offers rapid drug absorption [6,7]. Many topical dosage forms, such as solutions, tablets/lozenges, chewing gums, sprays, patches, films, hydrogels, hollow fibers, and microspheres, have been developed for oral mucosal delivery [8]. Among those, a mucoadhesive patch offers several unique advantages, such as direct and rapid onset, long retention time, sustained release, accurate dosing, and minimal unwanted side effects [4,9,10]. There are published studies reporting the in vivo application of mucoadhesive patches of certain drugs as a site-specific therapy to minimize systemic exposure [11,12,13]. In addition to the common buccal and sublingual routes, the supralingual (top of the tongue) route has also been investigated as a potential administration approach [14]. However, none of them has evaluated and compared the pharmacokinetic (PK) behaviors of drugs delivered via the supralingual route with the conventional intravenous (IV) and oral routes.

Mycophenolic acid (MPA) is a potent, selective, and reversible inhibitor of inosine monophosphate dehydrogenase (IMPDH) used in the prevention and treatment of organ transplant rejection in the clinic [15]. Recent studies showed that MPA also exerts a potent anticancer activity by inhibiting DNA synthesis and arresting the cell cycle in the S phase [16,17]. Interestingly, MPA-treated patients in the clinic seem to have a relatively lower risk of developing skin cancer, lymphoma, or lymphoproliferative malignancy compared to patients treated with other immunosuppressant agents [18,19]. Recently, Tsai et al. [20] found that MPA acts synergistically with different chemotherapeutic agents in killing various types of cancer cells, suggesting that MPA might be a potent chemo-adjuvant for the treatment of oral premalignant lesions. Currently, there are two oral forms of MPA available on the market, mycophenolate mofetil (MMF, CellCept^®^) and mycophenolate sodium (EC-MPS, Myfortic^®^) [21]. Oral administration of MPA is associated with several disadvantages, including adverse gastrointestinal effects and tolerability problems [22,23] and low drug levels at the targeted tissue due to limited blood supply around solid tumors [4]. To overcome these challenges, we explored the possibility of topical application of MPA via mucoadhesive patches.

There have been clinical PK studies of MPA conducted in patients with renal transplant [24,25], hematopoietic stem cell transplantation [26], childhood-onset systemic lupus erythematosus (cSLE) [27], and in a healthy Chinese population [28]. Chantal et al. described MPA plasma PK in renal transplant patients after twice-daily oral administration in a two-compartment model with zero-order absorption [24]. Payen et al. described MPA plasma PK in kidney transplant pediatric and adolescent patients using a two-compartment model with first-order absorption with a lag time [25]. Sherwin et al. and Zheng et al. described MPA and MPAG population PK using an enterohepatic recycling (EHR) model with different absorption compartments [26,27]. Colom et al. reported that the PK profile of orally administered MPA showed secondary/multiple peaks due to EHR through its inactive metabolite mycophenolic glucuronide (MPAG) [29]. Despite the fact that rich information can be found in the literature for establishing PK models and describing the EHR effect of MPA in humans, there is limited literature describing the compartmental PK for MPA in rats. Similar to human subjects, EHR also occurs in rats. However, different pharmacokinetics between humans and rats were found to be caused by the affinities of multidrug resistance-associated protein 2 (MRP2) for MPAG. Due to the affinity of rats MRP2 to MPAG being around 3.6 times more than human MRP2, MPAG is majorly excreted into the urine in humans and majorly excreted into bile in rats [30]. There were several pre-clinical PK studies of MPA in healthy rats [31,32,33] and Nagase analbuminemic rats [34]. Feturi et al. developed a semisolid topical formulation which was applied on the hind limb of rats, which resulted in 6% of bioavailability compared to IV administration and highly skin and muscle MPA accumulations [32]. Dridi et al. found circadian (light onset time) can largely influence the C_max_ and total exposure during MPA administration [33]. The half-life after 90 mg/kg single oral administration in rats was around 12.9 h [31]. Multiple peaks in PK profile after MPA administration were also observed in rats. However, no transmucosal delivery of MPA has been done yet. Therefore, PK evaluation of the supralingual route is needed.

Mucoadhesive patches applied directly on the lesion site can deliver drugs more effectively to the affected tissue compared to traditional oral administration. The incidence of the gastrointestinal side effect of MPA is associated with maximum plasma concentration (C_max_) after oral administration, and the relatively high fluctuation of the plasma concentration can increase the uncertainty of the amount of drug accumulated in the oral mucosa [35,36]. Supralingual administration of MPA may provide a continuous release of drug from the oral mucosa, which not only lowers the C_max_, but also reduces plasma concentration fluctuations. In this paper, we report and compare the PK and tongue tissue distribution studies of MPA in rats after IV, oral, and supralingual administration. The absolute bioavailability of oral and supralingual administration was calculated by comparing their mean areas under the curve (AUC_0–48_) to that of the IV administration. Different PK models were used to analyze different administration routes, and the most appropriate ones were selected for each of them. Multiple peaks were observed in the PK profiles of all three administration routes, and phase II metabolite MPA glucuronide was identified by LC-MS/MS in the plasma, which was usually responsible for EHR. The EHR model has been previously established for oral administration but never for transmucosal drug delivery of MPA. Our study reports a EHR model for supralingual PK analysis by an individual analysis method. The PK parameters indicate that the drug within the patch formulation had a sustained release from the dorsal tongue, resulting in low drug concentrations in the plasma and a high amount of the drug accumulated in the tongue tissue. This new route of supralingual administration has great potential as a valuable alternative approach for treating oral lesions.

## 2. Materials and Methods

### 2.1. Chemicals, Animals, and Instruments

MPA, griseofulvin (as an internal standard, IS), and formic acid were purchased from Sigma Aldrich (St. Louis, MO, USA). Mycophenolate sodium (MPA’s salt form) was purchased from USP (Rockville, MD, USA) and used for injection. Mucoadhesive patches containing MPA (3.6 µg/mm^2^) were designed and synthesized by Dr. Liu at the Baylor College of Dentistry, Dallas, TX, USA. Male Sprague Dawley (SD) rats were purchased from Envigo RMS (Indianapolis, IN, USA). An AB Sciex 4000 QTRAP^®^ UPLC-MS/MS system (Redwood City, CA, USA) was used to analyze all biological samples. LC-MS/MS grade water and acetonitrile purchased from VWR^TM^ Chemcial BDH^®^ (Chicago, IL, USA) were used as mobile phase for the quantification of MPA. Formic acid purchased from Sigma-Aldrich (St. Louis, MO, USA) was used to improve peak shapes of the UPLC-MS/MS method. Centrifuge Eppendorf 5417C tubes (Hauppauge, NY, USA) were used to separate plasma from whole blood and extract plasma from the organic solvent. Tissue homogenizer purchased from BioSpec Products, Inc. (Bartlesville, OK, USA) was used to get tongue homogenates. Phoenix WinNonlin Software (version 8.2.0.4383; Pharsight Corporation, Sunnyvale, CA, USA) was used to analyzing plasma PK parameters.

### 2.2. PK Studies and Tongue Tissue Distribution

Adult male Sprague-Dawley (SD) rats were used in all the animal studies. The animal experiments were approved by the Institutional Animal Care and Use Committee (IACUC) at Texas Southern University (TSU protocol #9080 approved on 30 May 2019) and were conducted according to the National Institute of Health “Guide for the Care and Use of Laboratory Animals, 8th Edition”. Rats were individually identified by tail markings and were acclimatized in the TSU vivarium for at least 7 days before animal experiments. Three groups of male SD rats (total *n* = 11, average weight of 379.8 ± 44.3 g) were used for PK studies of IV bolus injection, single oral administration, and supralingual patch application, respectively. Blood samples were taken at various times and centrifuged at 3000 rpm for 20 min to obtain plasma samples, which were then immediately stored at −80 °C until analysis. Two more groups of SD rats (total = 7, average weight 384.1 ± 64.9 g) were used for tongue tissue distribution studies with the same 4 h supralingual patch application time. No noticeable signs of discomfort were observed in any of the rats. Details of the experiments are described below.

#### 2.2.1. IV PK Study

Dosing solution was prepared by dissolving 1 mg of sodium MPA in 1 mL of normal saline and then filtered through a 0.22 µm filter (VWR^TM^, Chicago, IL, USA) before administration. The dosing solution was administered to the rat (*n* = 5) via the jugular vein as a bolus dose of 0.5 mg/kg of MPA. Blood samples (approximately 0.15 mL) were collected from the jugular vein before dosing (pre-dose) and at 2, 5, 15, 30 min, 1, 2, 3, 4, 6, 8, 10, 24, 28, 32, and 48 h post-dose.

#### 2.2.2. Oral PK Study

The same dosing solution as above was administered (0.5 mg/kg) to the rat (*n* = 3) by oral gavage with a 16-gauge feeding needle. Blood samples (~0.15 mL) were collected from the jugular vein at pre-dose, 5, 10, 15, 30 min 1, 2, 3, 4, 6, 8, 10, 12, 24, 28, 32 and 48 h post-dose.

#### 2.2.3. Patch PK Study

Polymeric mucoadhesive MPA patches (3.6 µg/mm^2^) were stored at −20 °C until use. In this study, a single dose patch (0.5 mg/kg, size around 50 mm^2^) was applied on the dorsal surface of the tongue while the rats were under light anesthesia, then was removed after 4 h (*n* = 3). The surface of the dorsal tongue was thoroughly washed with pure water for 5 min to remove the drug residual on the surface. Animals had free access to water and food after removing the patch. Blood samples (~0.15 mL) were collected from the jugular vein into heparinized centrifuge tubes at pre-dose, dose application period of 1, 2, 3, and 4 h (patch removed at this time point), and continued at 4.5, 5, 6, 7, 8, 9, 10, 24, 28, 32, and 48 h after the initiation of patch application.

#### 2.2.4. Tongue Tissue Distribution

Two groups of SD rats received patch application supralingually for 4 h the same way described in Section 2.2.3. The first group (*n* = 4) was sacrificed immediately (0 h), and tongues were thoroughly washed for 5 min and then removed for analysis. The second group (*n* = 3) was subjected to thorough tongue wash for 5 min at 0 h after patch removal and then sacrificed 20 h later, and tongues were removed for analysis.

### 2.3. Sample Preparation and Analysis

#### 2.3.1. Plasma Samples

Fifty (50) μL of rat plasma were extracted by adding 200 μL of acetonitrile containing 10 ng/mL of griseofulvin as the internal standard (IS) in a 1.5 mL centrifuge tube. The mixture was vortexed for 15 s and centrifuged at 14,000 rpm for 20 min at 4 °C. The supernatant was then injected into the LC-MS/MS instrument for quantitative analysis.

#### 2.3.2. Tongue Samples

Tongues collected from rats were washed 3 times with 1 mL of HPLC water, followed by adding 6 mL of water for every mass gram of tongue tissue and homogenization for 3 min. Tongue homogenates of 50 μL were extracted by adding 200 μL of acetonitrile with IS in a 1.5 mL centrifuge tube. The mixture was then vortexed for 15 sec and centrifuged at 14,000 rpm for 20 min at 4 °C. Five microliters of supernatant was then analyzed by the LC-MS/MS.

#### 2.3.3. LC-MS/MS Analysis

A sensitive and fast LC-MS/MS method previously developed and validated by our group [37,38] was used to measure the MPA amount. Both intra- and inter-day were within ±15% by calculation their coefficient of variation in rat plasma and tongue quality control samples [37]. Briefly, MPA separation was carried out on an ACE Excel 2 Super C_18_ column (50 × 2.1 mm, 2 μm) with a mobile phase run in the gradient elution of 0.1% formic acid in water (solvent A) and 0.1% formic acid in acetonitrile (solvent B) at a flow rate of 0.4 mL/min. Griseofulvin was used as IS. LC-MS/MS analysis was carried out on a 4000 QTRAP LC-MS/MS system with a Turbo Ion Spray ion source. Tandem mass spectrometry was employed under positive electrospray ionization to detect the specific precursor to product ion transitions *m*/*z* 321.2 → 207.2 for MPA and *m*/*z* 353.2 → 285.1 for the IS. The linear response ranged from 0.5 ng/mL to 1000 ng/mL (r^2^ > 0.999) for both the plasma and the tongue.

### 2.4. PK Model and Statistical Analyses

Phoenix WinNonlin Software and Phoenix^®^ NLME^TM^ Software (version 8.2; Certara L.P. Pharsight, St. Louis, MO, USA) were used to determine the PK parameters in the EHR model analysis. Preparation, exploration, and visualization of the data were performed using the Phoenix and GraphPad Prism (version 6.02; GraphPad Software, Inc., San Diego, CA, USA).

Various compartmental PK models were constructed to assess the PK of MPA for IV, oral and supralingual administration, respectively. To determine the most suitable compartmental model, we fitted data for MPA three- or four-compartment EHR model, with different combinations of the absorption description. The best and final model was chosen based on the lowest Akaike Information Criterion (AIC) values, best fitted predictive plasma concentrations, and least bias of the other diagnostic plots. Finally, all three types of administration were described by EHR PK models. A best-fit EHR model including central, peripheral, bile (A_bile_), and intestinal (A_gut_) compartments was used to describe the MPA plasma PK after IV (Figure 1a) and oral administrations (Figure 1b). A best-fit EHR model containing central, bile, and intestinal compartments were used for supralingual administration (Figure 1c). First-order kinetics was assumed for all PK processes other than absorption and bile excretion. Excretion from bile to the gut was described by zero-order kinetics. The rate of zero-order kinetics from the bile to gut was defined using Equation (1):GBr = 1/Tau,(1)
where Tau is the time interval for the bile emptying. GBr is the zero-order input from bile to gut.

Various compartmental PK models were constructed to assess the PK of MPA for IV, oral and supralingual administration, respectively. To determine the most suitable compartmental model, we fitted MPA data using three- or four-compartment EHR models. In the meantime, various absorption parameters were evaluated to find the one that best fit the absorption behavior of MPA. We first tried using first-order absorption but failed because of near-zero clearance estimates. A transit absorption model had a large bias on diagnostic plots and higher AIC. Finally, we used a combined transit and first-order absorption model to describe the absorption process, which gave the best fitting profile for our data. The chain of transition compartments (k_tr_) were used to describe the gradual and variable onset of drug absorption between the depot compartment and the central compartment, while ka_1_ and ka_2_ describe the fast and slow transition from the depot compartment to the central compartment and transition compartment, respectively. The number of transit compartments was optimized to 5 for both oral and supralingual absorption. k_tr_ is the transit rate constant from the first compartment to the fifth compartment (Figure 1b). The naïve pool estimate method was used for the PK model development and data fitting. For intra-individual variability, a proportional error model was used (Equation (2)):C_observed,ij_ = C_pred,ij_ × (1 + ε_ij_),(2)
where C_observed,ij_ is the observed plasma concentration in the ith individual at the jth time point, C_pred,ij_ is the predicted concentration, and ε_ij_ is the proportional residual error term under the assumption that ε *~* N(0,σ^2^), where the error term with mean of 0 and *σ^2^* is the variance assumed.

AIC is an extension of the minus twice the log-likelihood (-2LL) and the visual inspection of goodness-of-fit diagnostic plots. The reduction of AIC ≥ 2 was regarded as significant.

The area under the curve (AUC_0–48_) was calculated based on the trapezoidal rule manually. The mean estimated absolute bioavailability (F_ab_%) of the supralingual and oral administrations to the IV administration was calculated using Equation (3):F_abs_ (%) = [Mean AUC_(po/supralingual)_ × Dose_iv_]/[Mean AUC_iv_ × Dose_(po/suprealingual)_] × 100%,(3)

One-way analysis of variance (ANOVA) with post-hoc Turkey HSD (honestly significant difference) was used to determine the statistical significance of PK parameters among IV, oral and supralingual administrations by R Studio. Differences of *p* < 0.05 were considered significant for all statistical analyses.

## 3. Results

There have been limited studies on the potential use of a patch formulation for supralingual administration. This study described the PK and tongue distribution of MPA through a novel supralingual delivery route. The primary objectives of the current study were to develop PK models, to estimate the PK parameters and bioavailability of the patch formulation in rats compared to IV and oral administrations, as well as to evaluate the tongue tissue distribution after supralingual drug application. The results are presented below.

### 3.1. Estimated PK Parameters

The plots of the mean plasma concentration versus time profiles following IV, oral and supralingual administrations of MPA are shown in Figure 2. All plots display second/multiple peaks. Following IV administration, plasma MPA concentration showed an extra peak at ~3 h post-dose. For oral administration, MPA plasma concentration reached its first peak rapidly after 5 min, followed by a second peak also at ~3 h post-dose. For patch administration, trace concentration of MPA in plasma became detectable at 1 h and reached its first peak at ~4.5 h post-dose, and another broad peak was shown around 24–32 h. The plasma concentrations of MPA via IV and oral administrations declined quickly, while those via supralingual administration remained stable over time. The plasma concentration of MPA via the supralingual route remained very low but consistent, suggesting a limited and sustained drug release from the tongue tissue to the systemic circulation. The plasma AUC_0–48_ of MPA calculated by trapezoidal rule were 2170 ± 355 ng × h/mL, 1570 ± 218 ng × h/mL, and 165 ± 21.0 ng × h/mL, oral and supralingual administrations, respectively. The mean MPA elimination half-life value after IV and oral administration were 10.5 and 7.4 h. The half-life was derived from the last three time points of observation using WinNonlin by NCA analysis. Terminal half-life directly from the last three time points are not an appropriate indicator to describe MPA’s PK behavior due to the EHR of MPA; the real half-life will be longer than what we measured by NCA [39]. The values of absolute bioavailability of MPA were approximately 72.4 and 7.6% for the oral and supralingual doses, respectively. The mean central compartment clearance value after IV, oral and supralingual administration were 117 and 132, and 250 mg/kg×h, respectively. The mean inter-compartment apparent clearance value after IV and oral were 224 and 274 mg/kg × h, respectively. The EHR% decreased from supralingual administration (96.6%) to oral administration (69.2%) to IV administration (54.1%). Final parameters (mean ± standard deviation) are summarized in Table 1. There were no significant changes among half-lives, central compartment clearance, re-absorption rate constant from gut to the central compartment (K_gc_) for the three groups. As expected, the duration time from bile to gut (T_au_) after supralingual administration showed significantly longer than those from oral and IV administration. The central compartment volume of distribution (V/F) of MPA was also significantly larger than those from the other two groups, and the plasma-time area under the curve (AUC_0–48_) was significantly smaller than those from the other two groups. Even though the Ka_1_ and Ka_2_ of supralingual administration were larger than those from the oral administration, no statistical significance was established. K_tr_ (transit compartment absorption rate constant) of supralingual administration was significantly slower than that from an oral administration. The oral administration group also showed a faster rate constant from central to bile compartments.

### 3.2. Individual EHR PK Modeling Analysis

The graphic models that best described the plasma PK of MPA after IV, oral, and supralingual administration are shown in Figure 1. All of them recapitulated the multiple peak phenomenon of MPA in the plasma. It was assumed that MPA is metabolized to MPA glucuronide (MPAG) in the liver before excretion into the bile [40]. Excreted MPAG is then reconverted into the parent MPA by β-glucuronidase in the GI tract and re-absorbed into the systemic circulation. Therefore, four-compartment EHR models containing the central (MPA), peripheral (MPA), bile (A_bile_), and intestinal (A_gut_) compartment were used to describe the plasma PK of MPA after IV and oral administrations. A three-compartment EHR model containing the central, bile, and intestinal compartments was used for supralingual administration. Since the plasma concentration has less fluctuation after the absorption phase, adding the peripheral compartment to the model did not improve the fitting of the supralingual PK profile. Different approaches have been evaluated to describe the absorption process for the oral and supralingual administration, including first-order absorption without transit compartment, transit compartment only, and combinations of transit compartment and first-order absorption with lag time or without lag time. Based on the goodness-of-fit of plot and mean AIC values, the combined first-order absorption with transit compartment model without lag time was chosen to best describe oral administration absorption, and the combined first-order absorption with transit compartment model with a 0.5 h lag time was chosen to describe the supralingual absorption.

### 3.3. Model Evaluation

Diagnostic scatter plots from the Phoenix NLME program show the performance of the final EHR PK models by comparing observed concentrations versus individual predicted concentrations (Figure 3); individual weighted residuals versus individual predicted concentration and time after dose (Figure 4); observed concentrations and individual predicted concentration versus time after dose (Figure 5).

Figure 3 shows the comparison between the observed concentration versus individual predicted concentration for IV, oral and supralingual administration. The scatter plot shows that data are aligned on the line of unity across the entire range, indicating a good correlation between the observed and model-fitted values. Figure 4 shows the plots of the individual weighted residuals versus the individual predicted concentration (a, b, and c) or time after dose (d, e, and f) model for the three administration routes. The overall blue trend line lies around the line Y = 0, and the distribution is relatively flat across all time points along the horizontal red lines. All weighted residuals fell within the range of −2 to 2 of individual weighted residuals, suggesting that the proportional error model was appropriate and there was no major bias in the structural model. Figure 5 shows the goodness-of-fit relationship between the individual predicted concentration and observed concentration versus time after dose. Residuals are also uniformly distributed with time and MPA concentrations, and individuals fit well for the most part. The AIC were −80.5 ± 8.9, −112.3 ± 37.4, and −195.7 ± 4.9 for the IV, oral, and supralingual PK models, respectively. The mean AIC values were −173.3 ± 9.7 for supralingual administration if an additional peripheral compartment was added, −67.2 ± 10.4 for oral administration if using first-order absorption followed by two-compartment analysis, and −50.5 ± 26.2 for IV administration without describing the EHR. Since lower AIC means a better choice of model, we finalized the model based on both AIC values and goodness-of-fit of the plots.

### 3.4. Tongue Distribution

MPA’s tongue tissue distributions and patch residue are shown in Table 2. Compared to Figure 1, these data indicate that the concentration of MPA in the tongue tissue (42.8 μg/g at the 0 h and 0.8 μg/g after 20 h of removal the patch) was much higher than the highest concentration (<10 ng/mL) of MPA in the plasma. The result also shows that around 20.6% of the MPA still remained in the patch after 4 h of adhesion. Therefore, the actual dose was estimated to be around 0.4 mg/kg (79.4% of the original 0.5 mg/kg dose) and, the dose-adjusted absolute bioavailability was 7.6%.

## 4. Discussion

### 4.1. PK Parameters and EHR Phenomenon

In this study, we collected data after IV, oral, and supralingual administrations of MPA and analyzed the PK behavior in rats using the Phoenix NLME program. The bioavailability of oral and supralingual administration was manually calculated by trapezoidal rule for MPA plasma concentration vs. time area under the curve (AUC). The mean absolute bioavailability after oral administration was 72.4% by comparing AUC from 0 to 48 hr after IV administration, which was comparable to a previously published study (72–93%) [41]. The mean bioavailability after supralingual administration was estimated to be 7.6% (with dose adjustment), which was slightly higher than semisolid lipoderm formulation for topical hind limb administration in rats (6%) [32]. The mean half-lives after non-compartment analysis were 10.5 and 7.4 h after IV and oral administrations, respectively. The last three points for supralingual administration still fluctuated, so the half-life had a bias, and the non-compartmental analysis (NCA) could not catch the real half-life for supralingual administration. It may be due to the flip-flop phenomena that occurred in supralingual administration, and the MPA entered the systemic circulation at a rate slower than its elimination rate from the body. The half-life of oral administration (7.4 h) was shorter than the previously published number (12.9 h). Reasons for this could be that the previous publication did not capture the real terminal half-life with only a maximum of 35 h sampling time (less than three times of half-life), or they used a different formulation [31]. The transition rate constant (k_tr_) for the supralingual administration (0.193/h, Table 1) was much lower than that for the oral administration (1.25/h, Table 1), which may be caused by the slow diffusion rate of MPA released from the mucosal membrane into the systemic circulation. The chain of transition compartments (k_tr_) were used to describe the gradual and variable onset of drug absorption between the depot compartment and the central compartment, while ka_1_ and ka_2_ describe the fast and slow transition from the depot compartment to the central compartment and the transition compartment, respectively. In this study, five transition compartments were adequate to describe the PK model after oral and supralingual administration. The ka_1_ and ka_2_ of supralingual administration were higher than those of oral administration, maybe because part of MPA was quickly absorbed into the systemic circulation. On the other hand, the k_tr_ of supralingual administration was lower than that of oral administration, maybe because the tongue was acting as a reservoir for the rest of the drug for further sustained release.

### 4.2. EHR and PK Model Development

Secondary or multiple peaks have been reported in several pre-clinical and clinical PK studies of MPA [27,31,33,42], which showed that MPA is metabolized to inactive MPA glucuronide (MPAG) and pharmacologically active MPA acyl glucuronide (AcMPAG) by UGT1A9 and 2B7 in the liver, respectively [43] and that the concentration of MPAG is 20- to 100-fold higher than that of MPA in the plasma. MPAG is excreted into the bile by MRP2 and deconjugated back to MPA by gut bacteria and re-absorbed mainly in the proximal colon as MPA [44,45]. EHR can increase the area under the plasma concentration-time curve (AUC) of MPA by approximately 40% [29]. The exposure (AUC) inter-subject variability (>40%) after oral administration of MPA was also high, especially in early post-transplant patients [46,47]. Due to the affinity of MRP2 in rats being higher than in humans, more MPAG excreted through bile in rats while more MPAG excreted through urine in humans [30]. Evidently, our study also found that the PK of MPA was considerably affected by the EHR process. Therefore, we integrated the EHR into the PK models for the IV and oral administration of MPA first and then created the PK model for the supralingual administration. After establishing and comparing different combinations of PK models, we found that it was the best fitting to apply the four-compartment EHR model (central, peripheral, bile, and gut compartment) to describe the PK of IV and oral MPA administration (central, peripheral, bile, and gut compartment), whereas the PK of supralingual MPA administration was best described by the three-compartment EHR model (central, bile and gut compartment). The inter-compartment clearance (CL2) after IV and oral administration was higher than the central compartment clearance (CL), indicating that the peripheral compartment may not be a rate-limiting compartment as compared to the central compartment (Table 1). Thus, the peripheral compartment may be combined with the central compartment when fitting the PK model for supralingual administration. The EHR% via supralingual administration is higher than the other two routes of administration, causing the drug to stay longer in the system and take longer to get eliminated. The high EHR% also makes the PK profile more flat compared to those of IV and oral administrations.

Previously, several studies have been reported using different models to fit absorption phase after oral administration [28,29,40,48], including the transit compartment model with first-order absorption [49], short lag-time combined with long lag-time first-order absorption [50], first-order absorption [51], time-lagged first-order absorption [52], gamma distribution for absorption [44], and zero-order absorption [53]. After our evaluation of all those models, we found the transition compartment model with first-order absorption was the best one to describe our oral administration PK, and this method can also be used to fit supralingual PK absorption by adding a lag-time parameter with modification. Because the patch was applied on the dorsal tongue for 4 h, a model of infusion input with 0.5 h lag-time following transition compartment (first-order absorption) was selected to describe the supralingual absorption. The lag-time was 0.5 h since the plasma concentration was unable to be detected until 0.5 h after patch application, and the peak plasma concentration occurred at 0.5 h after the patch was removed. The transit model is an erlang/gamma-distributed delay approach for modeling delayed outcomes in PK [54,55], which can also be used for drugs with EHR or large molecules with absorption delay. We developed the two-process absorption model and used the transit compartment model combined with first-order absorption to pursue more mechanistic (semi-mechanistic) PK modeling approaches compared with typical absorption models. Usually, the transit compartment describes drug transit through a chain of identical compartments that are linked to the central compartment by a first-order absorption process [56]. In our study, the number of transit compartments was tested from one to ten and was then fixed at five based on the fittings of the PK profile.

The final EHR models had lower AIC values and better PK profile fitting compared to other types of PK models. Our PK results indicate that supralingual administration of MPA had lower exposure and less plasma concentration than oral and IV administration. It is critical to have a lower maximum plasma concentration after administration to reduce systemic side effects. Even though the plasma concentration was very low after supralingual administration, EHR phenomena occurred with multiple peaks shown in PK profiles. The chance of gastrointestinal toxicity by EHR after supralingual administration was lower compared to oral and IV administration due to its less fluctuation and lower plasma concentration. In summary, the PK models that we developed to describe the MPA disposition after IV, oral, and supralingual administrations support MPA exposure and EHR phenomena, which may be associated with some toxicity issues observed with other MPA formulations. Our PK-EHR models fitted well after IV, oral, or sublingual administrations of MPA, a drug that exhibits significant enterohepatic recirculation. The EHR model may be suitable for other drugs with EHR nature. Our PK-EHR model is also unique in demonstrating a good prediction capability for the prolonged release of MPA from the tongue after the patch application. Future studies are warranted to test the suitability of our PK-EHR model in other novel drug delivery systems and/or sustained release dosage formulations

### 4.3. Tongue Tissue Distribution after Supralingal Application

Despite the similar histology of oral mucosa and skin, oral mucosa is always moist by saliva, making it more permeable than skin, which is covered by a dry surface coated with sebaceous lipid [57]. The rank order of permeability in different parts of the oral mucosa in the oral cavity is determined by their relative thickness and degree of keratinization [58]. Buccal and sublingual are the most common routes for oral mucosa administration due to the absence of keratin, whereas the dorsal tongue surface is covered by a special mucosa consisting of both keratinized and nonkeratinized epithelium. Thus, supralingual drug delivery is expected to show a slower absorption and a lower plasma concentration compared to the sublingual and buccal administrations [59,60,61]. In addition, we found around 20.6% of MPA still remained in the patch after 4 h of adhesion, which means only less than 80% of the dose was released from the patch over the application period. The distribution equilibrium and contact time can influence bioavailability, especially when the doses are above the saturation solubility in the mucosa [6]. Those reasons above might lead to the low bioavailability of supralingual delivery of MPA.

The tongue biodistribution study showed that the amount of MPA in the tongue tissue is much higher than that in the blood at the same post-dose time points. It may be due to the mucosal membrane acting as a storage compartment for the drug once the drug is absorbed. The drug is stored in the mucosal membrane then slowly diffuses out into the systemic circulation [6], resulting in a sustained drug release, which is beneficial for controlling the therapeutic window. The accumulation of MPA at the site of the administration indicates that the supralingual patch may be a good approach for site-specific drug delivery for oral lesions. In the absence of measured MAP concentrations in tongue tissues following IV or oral administration, one cannot make a direct comparison with that of supralingual treatment. However, the observed higher MAP concentrations in tongue tissue over a prolonged period after patch removal is an indication of a good treatment outcome. It was reported that MPA concentrations in the skin, muscle, and draining lymph nodes (DLN) were much lower than that of in plasma after an IV administration. The skin-plasma ratio, muscle-plasma ratio, and DLN-plasma ratio were 0.085, 0.4, and 0.37 at 24 h after 10 mg/kg IV administration, respectively [32]. Thus, we assume that IV and oral administration would have minimum MAP accumulation in the tongue, and further studies are warranted to confirm the assumption. Studies of different dosing and various sites in the oral cavity, as well as the proof-of-concept anticancer efficacy of the novel MPA patch, are currently undergoing in our labs to provide further important information for future clinical application.

## 5. Conclusions

In summary, our study has established the PK parameters of MPA administered via supralingual mucoadhesive patch, IV injection, and oral administration. Our PK models were developed to describe their drug distribution profiles with multiple peaks of plasma concentration. Compared to IV and oral administration, the supralingual PK of MPA exhibited an atypical absorption profile, where MPA accumulated mostly inside of the mucosal membrane long after the removal of the patch. In addition, supralingual administration displayed much lower plasma concentrations of MPA compared to IV and oral dosing and showed slow partition from mucosal membrane to systemic circulation that may alleviate the peak concentration-related gastrointestinal side effects compared to oral administration. The results suggest the potential merits of higher accumulation of the drug at the site of administration by supralingual administration and a lower chance of gastrointestinal side effect. Further evaluation of its efficacy in pre-clinical animal models and humans is required.

## Figures and Tables

**Figure 1 pharmaceutics-13-00574-f001:**
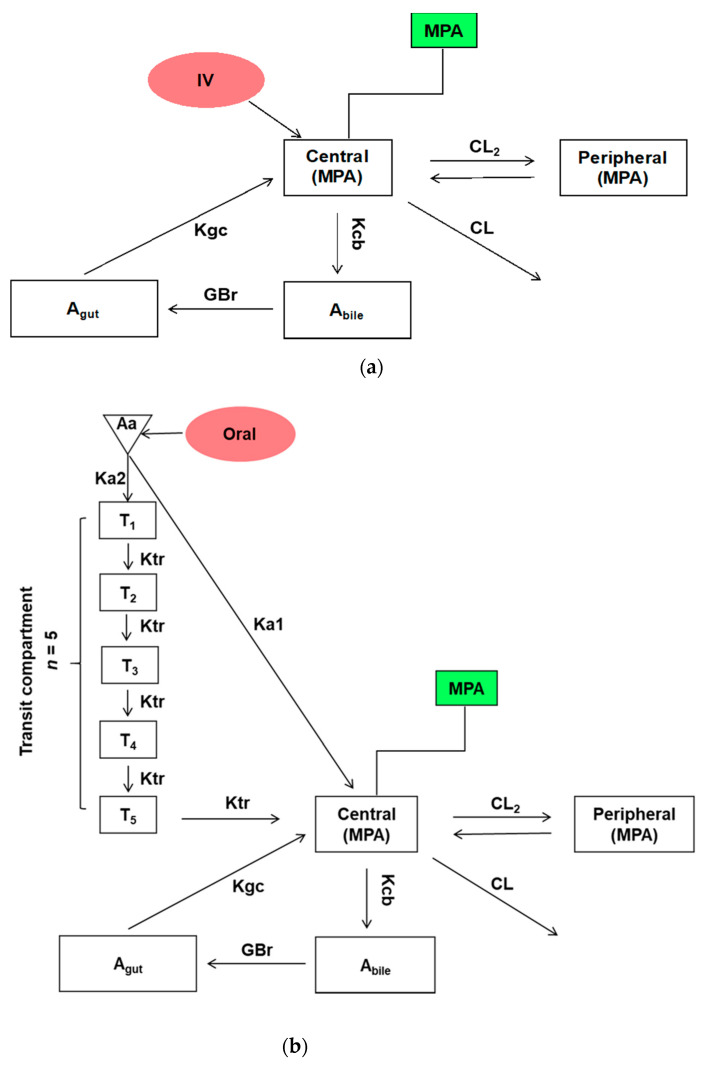
Structural models developed to analyze mycophenolic acid in rat plasma after three different routes of administration: (**a**) IV; (**b**) Oral; (**c**) supralingual administration. A_bile_: Amount in bile; A_gut_: Amount in the intestinal compartment. See Table 1 for the meaning of the different pharmacokinetic parameters.

**Figure 2 pharmaceutics-13-00574-f002:**
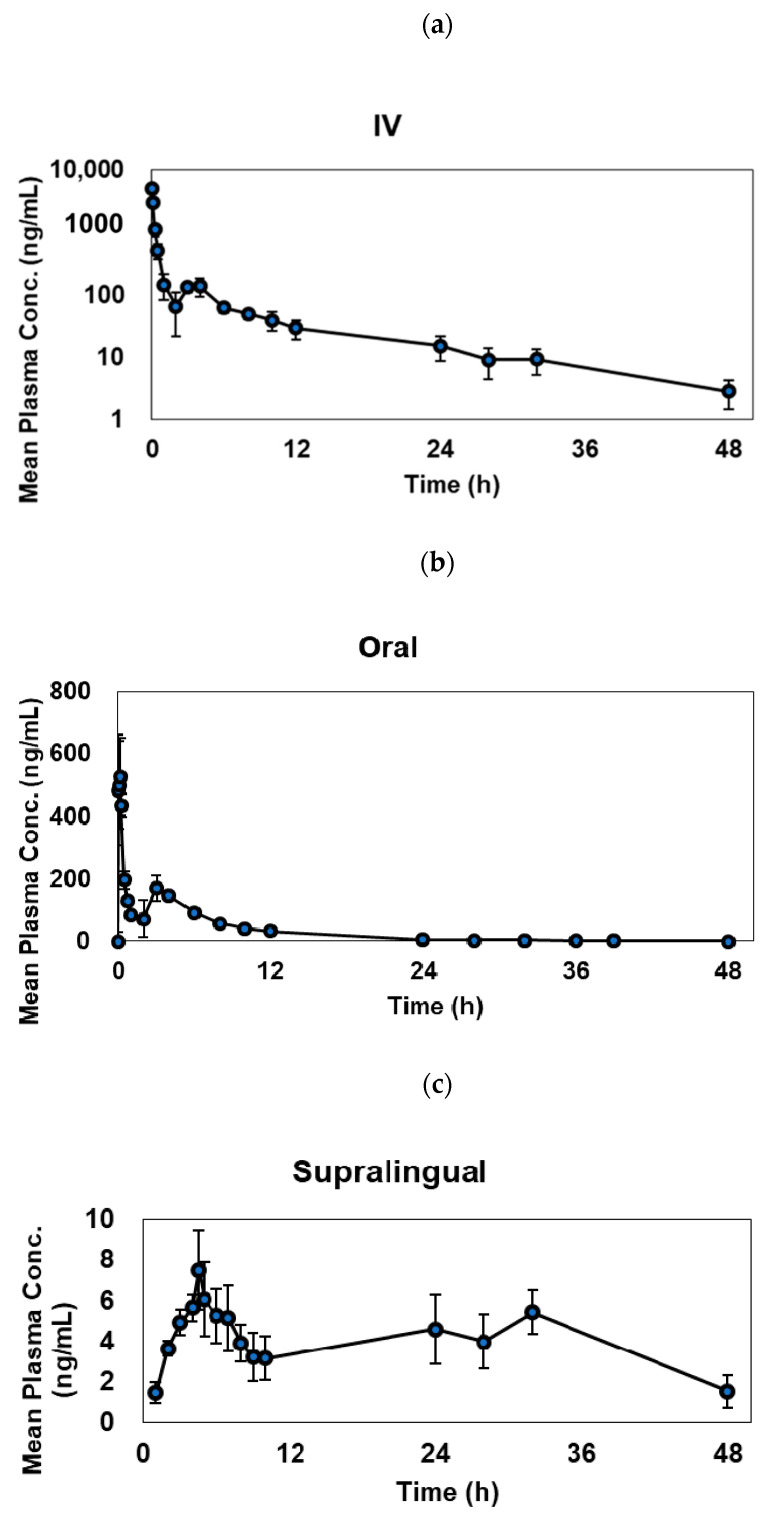
Average (mean ± standard deviation) pharmacokinetic profiles of mycophenolic acid after three different administrations: (**a**) IV (*n* = 5); (**b**) oral (*n* = 3); (**c**) supralingual (*n* = 3).

**Figure 3 pharmaceutics-13-00574-f003:**
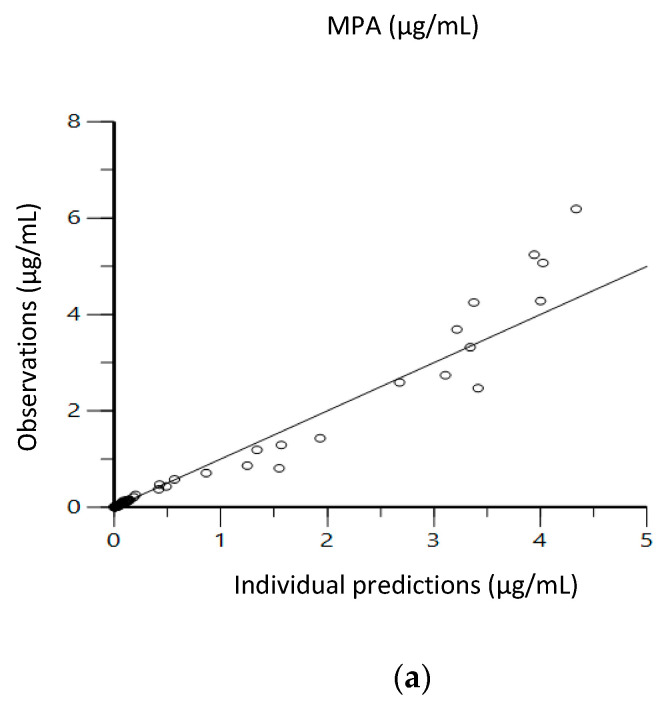
Goodness-of-fit plots showing observed vs. individual predicted mycophenolic acid plasma concentrations: (**a**) IV; (**b**) oral; (**c**) supralingual.

**Figure 4 pharmaceutics-13-00574-f004:**
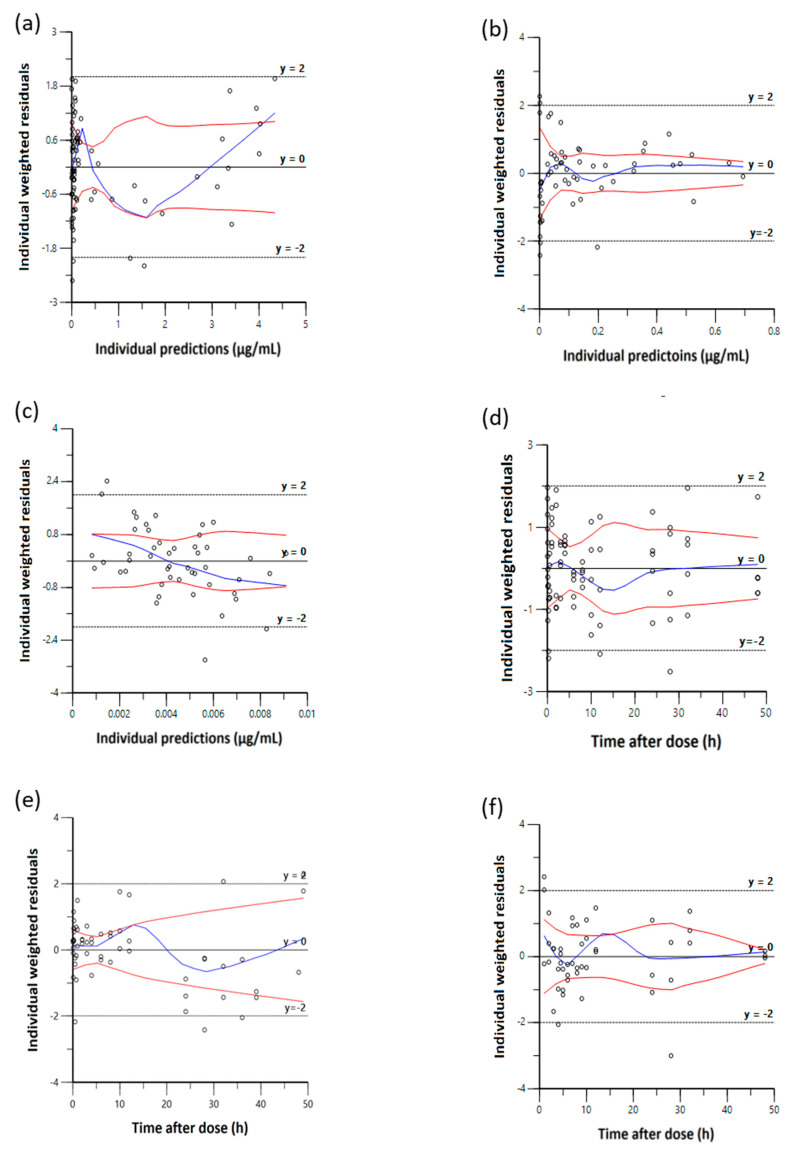
Goodness-of-fit plots showing individual predictions vs. individual weighted residuals of MPA: (**a**) IV; (**b**) oral; (**c**) supralingual. Individual weighted residuals vs. time after dose: (**d**) IV; (**e**) oral; (**f**) supralingual.

**Figure 5 pharmaceutics-13-00574-f005:**
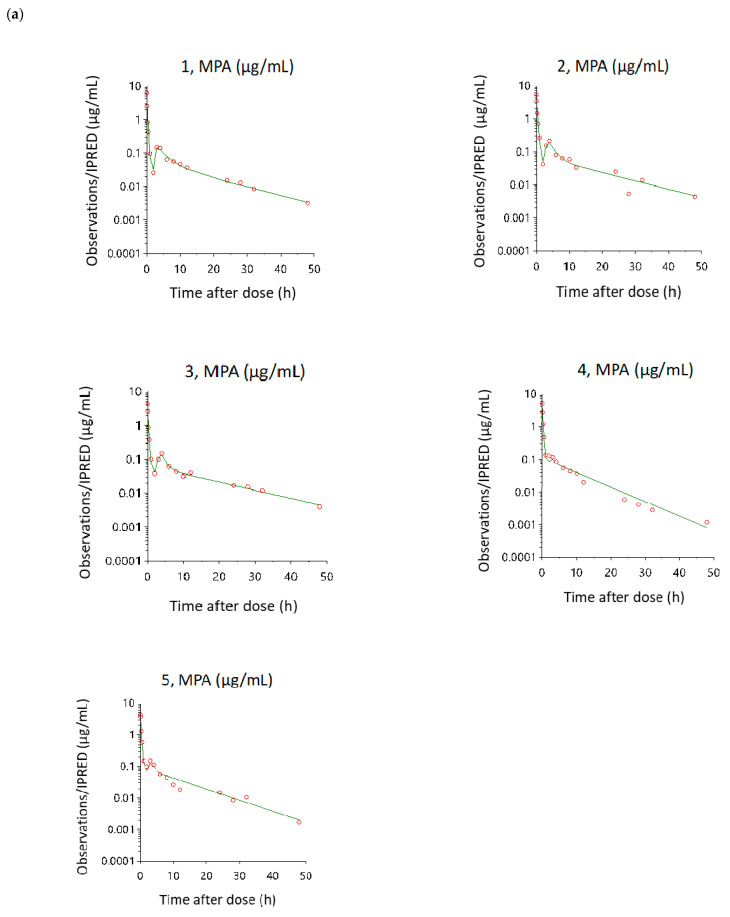
Goodness-of-fit plots of each rat showing individual predictions (IPRED, green line) or observations (red dots) versus time after dose route: (**a**) IV; (**b**) oral; (**c**) supralingual.

**Table 1 pharmaceutics-13-00574-t001:** Mean PK parameters after IV, oral, and supralingual administrations of MPA.

Parameter	Unit	IV (*n* = 5) Mean	Oral (*n* = 3) Mean	Supralingual (*n* = 3) Mean
Dose	mg/kg	0.5	0.5	0.5
Tau	hr	1.46 ± 0.535	4.54 ± 4.69	31.5 ± 11.6 ***,****
Half-life	hr	10.5 ± 1.20	7.40 ± 2.07	11.5 ± 2.98
CL/F_abs_	mL/(kg × h)	117 ± 92.2	132 ± 73.2	250 ± 333
CL_2_/F_abs_	mL/(kg × h)	224 ± 65.5	274 ± 162	NA
K_cb_	1/h	1.37 ± 1.02	10.3 ± 0.832 **,***	2.28 ± 3.62
K_gc_	1/h	1.97 ± 2.02	7.78 ± 8.52	32.4 ± 52.3
V/F_abs_	mL/kg	110 ± 10.8	29.3 ± 9.82	3090 ± 49.1 ***,****
V_2_/F_abs_	mL/kg	1740 ± 508	3000 ± 1690	NA
Ka_1_	1/h	NA	0.997 ± 0.313	21.6 ± 14.6
Ka_2_	1/h	NA	1.53 ± 0.366	37.3 ± 22.7
K_tr_	1/h	NA	1.25 ± 0.118	0.193 ± 0.040 ***
AUC_0–48_	ng × h/mL	2170 ± 355	1570 ± 218 **	165 ± 21.0 ***,****
F_abs_	%	NA	72.4	7.60
EHR	%	54.1 ± 10.3	69.2 ± 9.56	96.6 ± 49.5

Tau: duration of time goes from bile to gut using by zero-order input; CL/F_abs_: apparent clearance; CL_2_/F_abs_: apparent intercompartmental clearance; K_cb_: first-order rate constant from central compartment to bile compartment; K_gc_: re-absorption rate constant from gut to the central compartment; V_d_/F_abs_: volume of distribution of the central compartment; V_2_/F_abs_: volume of distribution of peripheral compartment; Ka_1_: fast-release depot that was readily absorbed into the plasma using a first-order rate constant; Ka_2_: slow-release depot that was readily absorbed into the plasma using a first-order rate constant; K_tr_: identical transfer rate constant of the transit compartment model using the first-order rate constant; AUC_0–48_: the area under the plasma drug concentration-time curve; F_abs_: absolute bioavailability. The EHR% was calculated according to the following equation: EHR% (IV and Oral) = K_cb_/(K_cb_ + CL/V +CL_2_/V_2_); EHR% (Supralingual) = K_cb_/(K_cb_ + CL/V). F_abs_ for supralingual administration was calculated based on the original dose subtract MPA patch residue (Table 2). ** Significant differences between IV and oral group. *** Significant differences between oral and supralingual group. **** Significant difference between the IV and supralingual groups.

**Table 2 pharmaceutics-13-00574-t002:** MPA concentrations stored in tongue tissue and remaining in patches.

MPA Concentration in Tongue Tissue after Patch Removal *	MPA Concentration	Percentage to Dose
0 h (*n* = 4)	42.8 ± 10.0 (µg/g)	3.8 ± 0.6%
20 h (*n* = 3)	0.8 ± 0.6 (µg/g)	0.11 ± 0.09%
Patch MPA residue ** (*n* = 7)	34.4 ± 5.2 µg	20.6 ± 2.8%

* Patches (0.5 mg/kg) were supralingually applied to both groups of rats for 4 h, then removed. The first group of rats (*n* = 4) were sacrificed immediately (0 h), and tongues were thoroughly washed for 5 min and then removed for analysis. The second group of rats (*n* = 3) were sacrificed 20 h later, and tongues were thoroughly washed for 5 min before removal for analysis. ** Patches collected from the two groups of rats above after 4 h of supralingual application were subject to analysis. This number was also used for supralingual F_abs_ calculation in Table 1.

## Data Availability

Not applicable.

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
