# Peer review of "Pharmacokinetic Model Analysis of Supralingual, Oral and Intravenous Deliveries of Mycophenolic Acid"

_pharmaceutics, 2021, doi:10.3390/pharmaceutics13040574_

Round 1

Reviewer 1 Report

In the present study, the authors have established a novel pharmacokinetic (PK) model for enterohepatic recycling after 3 different ways of treatment with mycophenolic acid (MPA) which is a potential candidate as anti-cancer medicine. They evaluated and compared those pharmacokinetics and concluded that supralingual administration of MPA using mucoadhesive patch for oral cancers such as oral squamous cell carcinoma because of constantly lower circulation level, compared with intravenous or oral treatment.

The parameters calculated by the model fits well with the measured ones and the results are interested and promising for clinical application.

I have only a few concerns as below;

  1. The authors concluded that the higher accumulation of MPA at the tongue is an advantage of supralingual treatment. But how about the MPA levels in tongue after the other 2 treatment methods?

  1. it would be better to show the four-compartment model, not only three- compartment model, of supralingual treatment. 

  1. Typos;

     Line 19: supraingual

     Line 20: supraglingual

Author Response

Reviewer #1: In the present study, the authors have established a novel pharmacokinetic (PK) model for enterohepatic recycling after 3 different ways of treatment with mycophenolic acid (MPA) which is a potential candidate as anti-cancer medicine. They evaluated and compared those pharmacokinetics and concluded that supralingual administration of MPA using mucoadhesive patch for oral cancers such as oral squamous cell carcinoma because of constantly lower circulation level, compared with intravenous or oral treatment. The parameters calculated by the model fits well with the measured ones and the results are interested and promising for clinical application.

COMMENTS: The authors concluded that the higher accumulation of MPA at the tongue is an advantage of supralingual treatment. But how about the MPA levels in tongue after the other 2 treatment methods?

RESPONSE:  We thank the Reviewer for raising this good question and have revised our conclusion accordingly. We agree that in the absence of measured MAP concentrations in tongue tissues following IV or oral administration, one cannot make a direct comparison with that of supralingual treatment. However, the observed higher MAP concentrations in tongue tissue over prolonged period after patch removal is an indication of a good treatment outcome. It was reported that MPA concentrations in skin, muscle, and draining lymph nodes (DLN) were much lower than that of in plasma after an IV administration. The skin-plasma ratio, muscle-plasma ratio, and DLN-plasma ratio were 0.085, 0.4, and 0.37 at 24hr after 10 mg/kg IV administration, respectively [Feturi et al 2018]. Thus, we assume minimum MAP accumulation in the tongue and further studies are warranted to confirm the assumption. The manuscript has been revised in Section 4.3 to reflect our discussion.

COMMENTS: It would be better to show the four-compartment model, not only three- compartment model, of supralingual treatment.

RESPONSE: Thanks for the Reviewer’s suggestion. We had tried a four-compartment model in our analytical model development process. We started with build-in PK one- and two-compartment models from Phoenix software then derived to three- and four-compartment EHR model. We found that a three-compartment EHR model had a better model fitness value of mean AIC as compared with a four-compartment EHR model (-195.7 versus -173.3).  Moreover, the four-compartment EHR model had more bias and weaker fitting than three-compartment EHR model by comparing their plasma concentration prediction vs. individual weighted residual diagnostic plots and individual predicted plasma concentration diagnostic plots, respectively. Thus, three-compartment EHR model was selected for data analysis. We have added this part to Section 2.4.

COMMENTS: Typos;

     Line 19: supraingual

     Line 20: supraglingual

RESPONSE: Thanks for Reviewer’s suggestion, we have revised the grammatical and formatting errors in the manuscript.

Reviewer 2 Report

The paper provides detailed data and models of the PK of a very interesting drug, MPA, after i.v., oral and supralingual application. The results are potentially important for the minimization of side effects and an effective, sustained local release at the site of desired action. The approach should also be of more general interest to a broader audience, well below the specific behavior of MPA. Overall, it is ratrher clear and well written.

What may deserve a more general description is the choice of the specific, fairly complex, multi-parameter models. The text moves from „various absorption models were evaluated“ on r 238 right to „The final model is..“ on r 239. It is trivial that the quality of a fit improves with the number of adjustable parameters, i.e., the criterion for choosing a model must be the progress in fit quality per parameter added or the failure of a simpler model to agree with the error ranges of the data. Given the substantial errors of the points in, for example, Fig 2a, which may not even unequivocally require a local minimum, one may wonder whether simpler models would still have provided a tolerable fit. Of course, the question of strictly required parameters (compartments) is essential for assigning a true meaning to the individual numbers.

The discussion also mentions the predictive power of the models for other dosing etc. – it would be interesting to elaborate on this a little more.

Numbers should be stripped to significant digits. For example, in Tab 1, 1739.3+/-508.0 mL/kg should read 1.7+/-0.5 L/kg or, at the most, 1.74+/-0.51 L/kg. And so on.

I think, each Fig. should fit on a single page.

Some sentences are lacking articles or do not really fit grammatically.

Author Response

Reviewer #2: The paper provides detailed data and models of the PK of a very interesting drug, MPA, after i.v., oral and supralingual application. The results are potentially important for the minimization of side effects and an effective, sustained local release at the site of desired action. The approach should also be of more general interest to a broader audience, well below the specific behavior of MPA. Overall, it is rather clear and well written.

COMMENTS: What may deserve a more general description is the choice of the specific, fairly complex, multi-parameter models. The text moves from „various absorption models were evaluated“ on r 238 right to „The final model is..“ on r 239. It is trivial that the quality of a fit improves with the number of adjustable parameters, i.e., the criterion for choosing a model must be the progress in fit quality per parameter added or the failure of a simpler model to agree with the error ranges of the data. Given the substantial errors of the points in, for example, Fig 2a, which may not even unequivocally require a local minimum, one may wonder whether simpler models would still have provided a tolerable fit. Of course, the question of strictly required parameters (compartments) is essential for assigning a true meaning to the individual numbers.

RESPONSE: We thank the Reviewer’s suggestion. We have revised the manuscript to include our model development process under Section 2.4:

“Various compartmental PK models were constructed to assess the PK of MPA for IV, oral and supralingual administration, respectively. To determine the most suitable compartmental model, we fitted MPA data using three- or four-compartment EHR model. In the meantime, various absorption parameters were evaluated to find the one that best fit the absorption behavior of MPA. We first tried first-order absorption, but failed because of near zero clearance estimates. A transit absorption model had large bias on diagnostic plots, and higher AIC. Finally, we used a combined transit and first-order absorption model to describe the absorption process, which gave the best fitting profile for our data.”

With the limited number of animals in our current study, it is however advisable for future studies to confirm that such multi-compartmental model analysis is the best suitable approach. 

COMMENTS: The discussion also mentions the predictive power of the models for other dosing etc. – it would be interesting to elaborate on this a little more.

RESPONSE: Thanks for Reviewer #2’s suggestion, we have revised accordingly as the following to last paragraph of Section 4.2: “Our PK-EHR models fitted well after IV, oral or supralingual administrations of MPA, a drug that exhibits significant enterohepatic recirculation. The EHR model may be suitable for other drugs with EHR nature. Our PK-EHR model is also unique in demonstrating a good prediction capability for the prolonged release of MPA from the tongue after the patch application. Future studies are warranted to test the suitability of our PK-EHR model in other novel drug delivery systems and/or sustained release dosage formulations.”       

COMMENTS: Numbers should be stripped to significant digits. For example, in Tab 1, 1739.3+/-508.0 mL/kg should read 1.7+/-0.5 L/kg or, at the most, 1.74+/-0.51 L/kg. And so on.

RESPONSE: Thanks for Reviewer #2’s suggestion. We changed all the numbers to three significant accordingly, but we would like to keep the unit as mL/kg because it is the preliminary study in rats.

COMMENTS: I think, each Fig. should fit on a single page.

RESPONSE: Thanks for Reviewer #2’s suggestion. We fit figures in Figure 2 and Figure 3 on a single page accordingly. Other figures were hard to change into the single page, we could wait for the suggestions from journal editor for further help.

COMMENTS: Some sentences are lacking articles or do not really fit grammatically.

RESPONSE: Thanks Reviewer #2’s suggestion, we have revised the grammatical and reference errors in the manuscript.